# The Risk of Venous Thromboembolism in Korean Patients with Breast Cancer: A Single-Center Experience

**DOI:** 10.3390/cancers15123124

**Published:** 2023-06-09

**Authors:** Jung Ho Park, So Eun Ahn, Lyo Min Kwon, Ho Hyun Ko, Sanghwa Kim, Yong Joon Suh, Ho Young Kim, Kyoung-Ha Park, Doyil Kim

**Affiliations:** 1Division of Breast and Endocrine Surgery, Hallym University Sacred Heart Hospital, Anyang 14068, Republic of Korea; ringri@hallym.or.kr (J.H.P.);; 2Department of Radiology, Hallym University Sacred Heart Hospital, Anyang 14068, Republic of Korea; 3Department of Thoracic and Cardiovascular Surgery, Hallym University Sacred Heart Hospital, Anyang 14068, Republic of Korea; xeros1024@hallym.or.kr; 4Division of Hematology-Oncology, Hallym University Sacred Heart Hospital, Anyang 14068, Republic of Korea; ksfamily@hallym.or.kr; 5Division of Cardiovascular Disease, Hallym University Sacred Heart Hospital, Anyang 14068, Republic of Korea; pkhmd@hallym.or.kr

**Keywords:** venous thromboembolism, breast neoplasms, Asian, risk factors

## Abstract

**Simple Summary:**

In Asian patients with breast cancer, venous thromboembolism (VTE) has been an under-recognized condition. We demonstrated close association between VTE and breast cancer in a Korean cohort. The incidence of VTE was not low, as previously described. The risk factors for VTE included old age, male breast cancer, chronic kidney disease, breast reconstruction, and advanced stage. VTE was associated with poor disease-free and overall survival. Most of the patients with VTE were manageable with anticoagulation, although mortality occurred in some patients. Our findings provide valuable information on comprehensive care for Asian patients with breast cancer.

**Abstract:**

The relationship between cancer and venous thromboembolism (VTE) has long been described. The risk of VTE in Asian patients with breast cancer remains largely unknown. This study described the incidence and risk factors of VTE in Korean patients with breast cancer. Data were collected from a retrospective database of patients who underwent breast cancer surgery between 2011 and 2020 at a single institution. The Cox proportional-hazards model was used to identify factors associated with VTE occurrences. Among the 2246 patients with breast cancer, 48 (2.1%) developed VTE during a median follow-up period of 53 months. The average incidence of VTE was 459 per 100,000 person-years. Age ≥ 60 years, male sex, chronic kidney disease, reconstructive procedures, and stage II or higher were independent predictive factors for VTE. VTE was associated with poor disease-free survival (hazard ratio (HR), 6.140; 95% confidence interval (CI), 3.480–10.835), and overall survival (HR, 8.842; 95% CI 4.386–17.824). Most VTE events were manageable with anticoagulation; three (6.3%) patients died of VTE, despite intensive care. The incidence of VTE was significantly elevated in Korean patients with breast cancer. Since VTE has a negative effect on oncologic outcomes of breast cancer, clinicians should manage its risk throughout their lifetime.

## 1. Introduction

Venous thromboembolism (VTE) is defined as a thrombotic obstruction, involving the venous system and pulmonary artery. The incidence of VTE in the general population was reported to be 100 per 100,000 person-years. The diagnosis of VTE is largely dependent on clinical suspicion. However, patients with VTE present non-specific symptoms, and diagnostic tests for VTE are suboptimal [1]. Because VTE is a cause of preventable death in hospitals, screening and prevention for high-risk patients are important.

The relationship between cancer and VTE has long been described [2]. Cancer can evoke the three components of Virchow’s triad: venous stasis, endothelial injury, and hypercoagulable state [3]. Specifically, venous stasis occurs during general anesthesia and cancer surgery. Further, endothelial injury can occur during chemotherapy and central venous access [4], and cancer induces a thrombogenic state by secreting procoagulant molecules [5]. Moreover, VTE is one of the major causes of mortality among patients with cancer [6].

The risk of VTE is closely associated with tumor biology. Previous studies of cancer-associated VTE have focused on high-risk malignancies. Cancers of the pancreas, lungs, and brain are high-risk factors for VTE, whereas breast cancer is at low risk for VTE [6]. Breast cancer is associated with a low risk of developing VTE, since the operation time is usually short, and the patients are fully ambulatory post-surgery. Therefore, VTE was not a major concern in patients with breast cancer. However, with advances in breast cancer treatment, the prevalence of breast cancer is increasing, and the risk of VTE is becoming more significant as the lifespan of breast cancer patients increases [3].

The incidence of VTE in the Asian population is lower than that in the Western population [6]. Additionally, VTE is an underrecognized condition in Asian countries [7,8], and thromboprophylaxis is not commonly utilized [9]. This disparity may be due to genetic differences in thrombophilia between Asian and Caucasian people [10]. Other than inherited conditions, cancer is the most common cause of VTE, and is responsible for 15–33% of VTE in Asian patients [11,12]. However, few studies have described the risk factors of VTE in Asian patients with breast cancer. Therefore, this study described the incidence and risk factors of VTE during multimodal treatment for breast cancer in a Korean cohort.

## 2. Materials and Methods

### 2.1. Data Collection

Data were collected from a retrospective database of patients who underwent breast cancer surgery at a single tertiary institution between January 2011 and December 2020. Consecutive patients diagnosed with breast cancer during this period were included. Exclusion criteria were patients who refused breast surgery, those with occult breast cancer, ongoing treatment for VTE, and a follow-up period of <1 month. Further, clinicopathological, radiological, and survival data were obtained.

### 2.2. Procedures

Patients were routinely examined using mammography and breast ultrasonography before surgery. When a suspicious lesion was identified using breast ultrasonography, a core-needle biopsy was performed for the lesion. When a lesion presented as microcalcifications on a mammogram and was not visible on an ultrasonography image, an excisional biopsy was performed after wire-guided localization.

Surgical procedures for breast cancer were performed in a standard manner. Breast-conserving surgery or total mastectomy was performed at the discretion of the surgeon. For patients with clinically negative axilla, sentinel lymph nodes were routinely sent for intra-operative frozen sectioning. For those with positive axillary lymph nodes, axillary lymph node dissection was routinely performed. During breast-conserving surgery, breast tissues were sampled from six directions and sent for intra-operative frozen sectioning to evaluate resection margins. An adequate safety margin was achieved by additional resection of the involved margin. If a negative resection margin was not achieved, a completion total mastectomy was performed.

Patients were treated with either anthracycline-based or taxane-based chemotherapy when indicated. Chemotherapeutic agents were routinely administered via a totally implantable vascular access device (i.e., chemo-port) at 3-week intervals. Neoadjuvant chemotherapy was mainly offered to patients with locally advanced breast cancer. For patients who underwent upfront surgery, adjuvant chemotherapy was planned after surgery. Adjuvant chemotherapy was usually recommended for tumors greater than 1 cm in size or when axillary lymph nodes were positive.

Adjuvant radiation therapy was offered after chemotherapy was over. For those who were not indicated for chemotherapy after surgery, radiation therapy was planned after at least 2 weeks. 

Adjuvant endocrine therapy was administered when the hormonal receptor were positive. For the patients with carcinoma in situ, tamoxifen was administered for 5 years. For premenopausal women with invasive carcinoma, goserelin was added for 2–5 years. For postmenopausal women with invasive carcinoma, an aromatase inhibitor was administered for 5–10 years.

Computed tomography scans of the chest and abdomen were performed at 3–6 months intervals to monitor breast cancer recurrence; however, systematic screening for VTE was not routinely performed. When a patient was suspected of having VTE, either computed tomography or ultrasonography of the lower extremity was performed.

### 2.3. Outcome Variables

VTE was classified into deep vein thrombosis (DVT), pulmonary thromboembolism (PTE), or both. DVT was classified as upper extremity DVT or lower extremity VTE. Thrombosis involving the axillary, subclavian, internal jugular veins, and right atrium was classified as upper extremity DVT. Further, thrombosis involving the iliac, femoral, popliteal, and tibial veins was classified as lower extremity DVT.

Disease-free survival was defined as the time interval between curative surgery for primary breast cancer and recurrence, overall survival was defined as the time interval between breast cancer diagnosis and in-hospital mortality or hospice discharge, and the VTE-free interval was defined as the time interval between breast cancer diagnosis and occurrence of VTE.

### 2.4. Statistical Analysis

Incidence rates of VTE were calculated based on person-years. Continuous variables were presented as median and range. Categorical variables were presented as frequencies and percentages. Comparisons between continuous variables were performed using the Mann–Whitney U test. Comparisons between categorical variables were performed using the chi-squared test. Survivals between groups were compared using log-rank tests. The Cox proportional-hazards model was used to identify factors associated with VTE occurrences. Statistical significance was set at *p <* 0.05. Statistical analyses were performed using the Statistical Package for Social Sciences (version 27.0; IBM Corp., Armonk, NY, USA).

## 3. Results

### 3.1. Clinicopathologic Characteristics of the Patients

A total of 2246 patients were included in the study (Figure 1). The clinicopathological characteristics of the patients are summarized in Table 1. The median age of the patients was 51 years (range, 21–91). Almost all patients (99.7%) were female. The median body mass index (BMI) was 23.9 kg/m^2^ (range, 14.1–45.4). The majority of the patients (72.9%) had stage I or II breast cancer. Histologic subtypes of breast cancer were mostly invasive ductal carcinoma (73.3%), followed by ductal carcinoma in situ or microinvasive carcinoma (17.2%), invasive lobular carcinoma (3.0%), mucinous carcinoma (2.2%), medullary carcinoma (0.7%), metaplastic carcinoma (0.7%), and others (2.8%). Hormonal receptor was positive for 76.0%, and HER2 was positive for 25.1%. The majority of the patients (74.8%) underwent breast-conserving surgery, 69.7% of the patients received chemotherapy, and 76.9% of the patients received endocrine treatment.

### 3.2. Factors Associated with VTE Occurrence

Patients who developed VTE were associated with old age, male sex, reconstructive procedures, advanced stage, and chemotherapy (Table 1). None of the patients with ductal carcinoma in situ developed VTE.

We further classified the comorbidities of patients into various disease categories (Appendix A). Patients with VTE were significantly more likely to have hypertension (47.9% vs. 23.8%, *p <* 0.001), diabetes mellitus (20.8% vs. 9.2%, *p =* 0.020), and chronic kidney disease (4.2% vs. 0.4%, *p =* 0.022). The prevalence of various comorbidities, including malignancy other than breast cancer, psychiatric disorder, asthma, chronic obstructive pulmonary disease, and previous history of stroke were more frequent among the patients with VTE. However, these associations were statistically not significant.

Factors associated with VTE occurrence were analyzed in Table 2. Univariable analysis revealed that age ≥ 60 years, male sex, chronic kidney disease, reconstructive procedures, stage II or higher, HER2 positivity, and chemotherapy were associated with VTE. On multivariable analysis, age ≥ 60 years, male sex, chronic kidney disease, reconstructive procedures, and stage II or higher were independent predictive factors for VTE.

### 3.3. Occurrence of VTE over Time

The occurrence of VTE was analyzed as a time-dependent variable. Forty-eight (2.1%) patients developed VTE during a median follow-up of 53 months (range, 0–140 months). The incidence of VTE was 459 cases per 100,000 person-years, on average. Among the 48 patients with VTE, 27 (56.3%) were diagnosed within 6 months of breast cancer diagnosis, 31 (64.6%) were diagnosed within 12 months, and 41 (85.4%) were diagnosed within 24 months (Figure 2). The incidences of VTE during the initial 6 months, 12 months, and 24 months of diagnosis were 2433, 1456, and 990 cases per 100,000 person-years, respectively. Median VTE-free survival was 4.5 months (range, 0–73 months). 

### 3.4. Oncologic Outcomes of the Patients with VTE

The association between VTE and the oncologic outcomes of patients is shown in Figure 3. Of the 2157 patients with primary breast cancer, there were 161 (7.5%) cases of recurrences and 71 (3.2%) cases of mortality during a median follow-up of 54 months (range, 0–140 months). VTE was associated with poor disease-free survival (hazard ratio (HR), 6.140; 95% confidence interval (CI), 3.480–10.835; *p <* 0.001) and poor overall survival (HR, 8.842; 95% CI, 4.386–17.824; *p <* 0.001). The occurrence of VTE was an independent predictive factor for recurrence (Table 3) and mortality (Table 4). 

### 3.5. Clinical Presentations of the Patients with VTE

The clinical characteristics of the patients with VTE are shown in Table 5. Twenty-five (52.1%) events were isolated PTE, 18 (37.5%) were isolated DVT, 5 (10.4%) were PTE combined with DVT. Among the 30 patients with PTE, 21 (70%) were asymptomatic. Among the 23 patients with DVT, 15 (65.2%) had lower extremity DVT, and 8 (34.8%) had upper extremity DVT. All patients with upper extremity DVT had a venous access device. All but one lower extremity DVT presented with lower extremity edema.

Of the 48 patients with VTE, 10 (20.8%) patients had a condition other than breast cancer. Five (10.4%) patients were associated with recent traumatic injury, three (6.3%) were suspected of May-Thurner syndrome, two (4.2%) were immobile, and one (2.1%) was diagnosed with essential thrombocytosis.

Most (79.2%) of the patients received anticoagulation for VTE. Warfarin was administered to 10 (20.8%) patients, with or without low-molecular-weight heparin. Direct oral anticoagulants were administered to 24 (50%) patients. Ten (20.8%) patients with VTE were observed without anticoagulation; nine (18.8%) spontaneously recovered; one (2.1%) developed subsequent atrial flutter. Although anticoagulation was successful in most of the patients with VTE, three (6.3%) patients died from VTE, despite intensive care.

There was no significant difference on clinicopathological characteristics between asymptomatic and symptomatic VTE (Appendix A). The factors associated with the occurrence of symptomatic VTE were nearly identical to those associated with all VTE (Appendix A). The effect of symptomatic VTE on recurrence and mortality was almost identical to that of all VTE (Appendix A). There was no significant difference in recurrence and mortality between asymptomatic and symptomatic VTE (Appendix A).

## 4. Discussion

In this retrospective study conducted at a single institution in Korea, the incidence of VTE was significantly elevated in patients with breast cancer. The incidence of VTE in the Korean population was reported to be approximately 42.2 per 100,000 person-years [11], which is less than half of that in the Western population. The average incidence of VTE in our study was 459 per 100,000 person-years, which is 10.9 times as high as that in the general population. Additionally, the cumulative incidence of VTE in Korean patients with breast cancer was 2.1% during 53 months of follow-up, which is comparable to previous data in Western studies [13,14]. However, it was considerably lower than that of advanced gastric cancer (3.5%) [15], pancreatic cancer (5.8%) [16], and non-small cell lung cancer (6.4%) [17] in Korea.

The incidence of VTE was highest right after initial diagnosis and gradually decreased over time. More than half of the VTE events occurred within 6 months of breast cancer diagnosis. The cumulative incidence of VTE at 6 months was 1.2%. This finding is consistent with previous studies, suggesting that the peridiagnostic period is vulnerable to VTE in patients with cancer [18,19]. Surgery and chemotherapy are initiated during this period, which further elevate the risk of VTE [3]. Considering that most of the patients included in this study were diagnosed with early breast cancer, thromboprophylaxis for VTE during the perioperative period is important.

For patients scheduled for general surgery, the perioperative risk of VTE can be evaluated, according to the Caprini score, which incorporates the patient’s age, type of surgery, medical condition, and family history [20,21]. According to the Caprini risk assessment model, cancer patients over 40 years of age are considered high risk, and anticoagulation with low-molecular-weight heparin is recommended for them. However, the Caprini score overestimates the risk of VTE in patients with breast cancer [22]. Most patients with breast cancer do not need pharmacologic prophylaxis during the perioperative period [22,23]. Nevertheless, we suggest anticoagulation for selected patients at high risk of developing VTE. Total mastectomy alone did not significantly increase the risk of VTE in our data; however, breast reconstruction was a strong risk factor for VTE. Many authors have also recommended anticoagulation for patients who are scheduled for breast reconstruction [24,25].

Although chemotherapy is a well-known risk factor for VTE, there is insufficient evidence to administer pharmacologic prophylaxis for most patients with breast cancer. According to Khorana et al., breast cancer is considered to be a low-risk cancer for VTE occurrence [26]. Most of the patients with breast cancer were fully ambulatory during chemotherapy, and did not need anticoagulation. Currently, clinical guidelines do not recommend anticoagulation for VTE prophylaxis during chemotherapy in patients with breast cancer [27,28]. When using anticoagulation, the risk of VTE should be balanced with the risk of bleeding. In patients with breast cancer without brain metastasis, the risk of major bleeding is minimal compared to those with gastrointestinal malignancy [29]. Therefore, we suggest liberal use of anticoagulation for a subset of patients at a high risk of VTE.

Old age is one of the most important factors for VTE occurrence. It is noteworthy that the age distribution of Korean patients with breast cancer is lower than that of Western patients. The incidence of breast cancer peaks at 45–49 years in Korean women, whereas the peak age is over 60 in Western countries [30,31]. The younger age at diagnosis further decreased the risk of VTE among Korean patients with breast cancer. Considering that the incidence of breast cancer and the proportion of elderly patients are increasing in Korea [31], the burden of VTE is also expected to increase in the future. 

Male breast cancer was a strong risk factor for VTE, although there were only 6 (0.3%) male patients in this study. Therefore, it is insufficient to conclude a positive association between VTE and male breast cancer. While one study showed only a 1.57-fold increased risk of VTE in male breast cancer [32], another suggested that male patients with breast cancer treated with tamoxifen are at high-risk for VTE [33]. As the number of male breast cancers is limited, the association between male breast cancer and VTE should be further investigated.

There were associations between VTE and various comorbidities. Previous studies have shown that cardiopulmonary diseases are associated with an increased risk of VTE [20]. Among various comorbidities, chronic kidney disease was particularly important for VTE in this study. It has been suggested that the increased risk of VTE is attributable to factor VIII activation in chronic kidney disease [34]. Treating this subgroup of patients is challenging in clinical practice, as they are at increased risk of VTE and are more vulnerable to hemorrhagic complications of anticoagulants [35].

We were unable to demonstrate that tamoxifen elevates the risk of VTE. Rather, tamoxifen was inversely associated with VTE. Because we prefer aromatase inhibitors for postmenopausal women, those who received tamoxifen were younger than those who received an aromatase inhibitor. A previous Asian study also reported no association between tamoxifen and VTE. In 28,029 Taiwanese patients with breast cancer, no difference in the incidence of VTE between the tamoxifen and non-tamoxifen groups was observed [36]. However, another recent study showed that tamoxifen increased the risk of VTE two-fold among Taiwanese patients aged > 65 years [37].

There are still other risk factors that also need to be considered during VTE risk assessment. Although obesity is an established risk factor for VTE [20], there was no significant association between BMI and VTE in the present study. The impact of obesity was not significant because the prevalence of obesity was low in Korean women during the study period. Considering that the prevalence of obesity is increasing in Korea [38], the association between BMI and VTE should be evaluated in future studies. Bilateral breast cancers prolong operation time and may lead to prolonged bed rest. Additionally, venous access to the lower extremity is occasionally needed [39]. Bilaterality did not elevate the risk of VTE in our study; however, clinicians should recognize this potential risk factor.

We demonstrated an association between VTE and negative oncologic outcomes of breast cancer. VTE was associated with a 6.14-fold and 8.84-fold increased risk of recurrence and mortality, respectively. VTE was an independent prognostic factor for recurrence and mortality. Previous studies also showed that VTE worsens the survival of patients with various cancers, including pancreatic cancer [40], lung cancer [17], stomach cancer [41], colorectal cancer [42], hematologic malignancy [43], ovarian cancer [44], endometrial cancer [45], and melanoma [46]. There are several explanations for this association. First, VTE can be a direct cause of mortality among cancer patients. It was the second-most common cause of mortality after breast cancer progression [2]. Second, the occurrence of VTE may indicate a high tumor burden. The presence of circulating tumor cells can activate the coagulation pathway [47]. Third, the occurrence of VTE may indicate aggressive characteristics of the tumor. In a recent study of cancer-associated VTE, somatic mutations of *KRAS* and *STK11* were associated with thromboembolic events [48,49]. However, mutations frequently identified in breast cancer, such as *TP53* mutations, did not elevate the risk of VTE [48,50].

There was a high proportion of upper extremity DVT; especially, one-third of DVT cases in our study were upper extremity DVT. Upper extremity DVT comprised only 5% of all DVT cases in a large series of DVT registries [51]. A high incidence of upper extremity DVT is associated with central venous access. We also observed a relatively larger number of PTE events than DVT events. Two-thirds of the patients with PTE were asymptomatic, which is related to incidental findings on computed tomography. Anticoagulation is usually recommended for patients incidentally diagnosed with VTE [52]. However, either watchful waiting or a short course of anticoagulation is acceptable when only a subsegmental artery is involved [27].

Our study has some limitations. The study was performed at a single center and had a retrospective design. Although we collected data from consecutive patients during the study period, this study had a selection bias. Since most patients who underwent surgery were ambulatory, patients with poor performance may be excluded from the study. As the number of patients with VTE was small, statistical power was not sufficient to validate the various risk factors for VTE. Nevertheless, our study is the first to identify risk factors of VTE in Asian patients with breast cancer in a time-dependent manner. Whether anticoagulation can reduce the incidence of VTE and the oncologic outcome of breast cancer should be further evaluated. 

## 5. Conclusions

The risk of VTE in Korean patients with breast cancer was comparable to that in Western countries. The incidence of VTE was highest at the time of diagnosis and declined over time. Additionally, the occurrence of VTE was associated with poor oncologic outcomes of breast cancer. Hence, clinicians should manage the risk of VTE throughout their lifetimes. Further studies are needed to compare the risk and benefits of anticoagulation for high-risk patients.

## Figures and Tables

**Figure 1 cancers-15-03124-f001:**
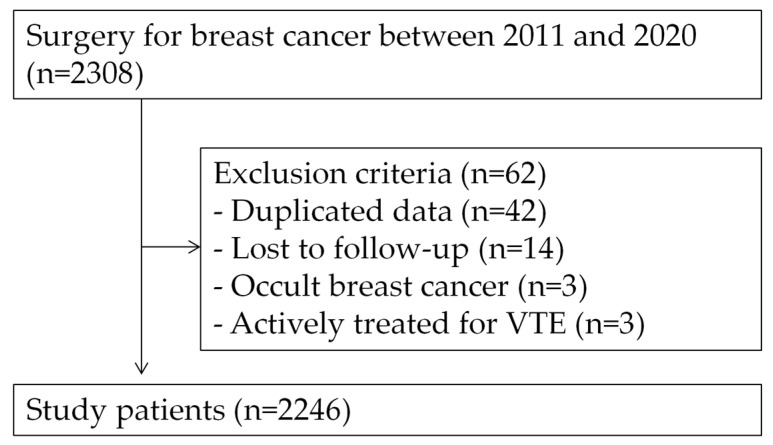
Flow diagram of the study design.

**Figure 2 cancers-15-03124-f002:**
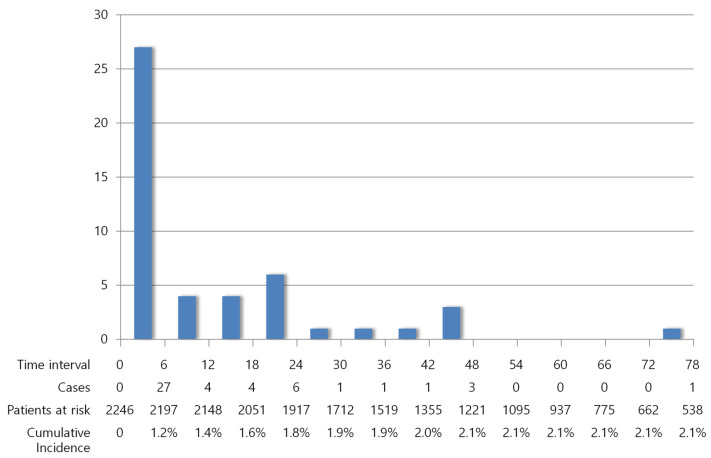
Decreasing incidence of VTE over time.

**Figure 3 cancers-15-03124-f003:**
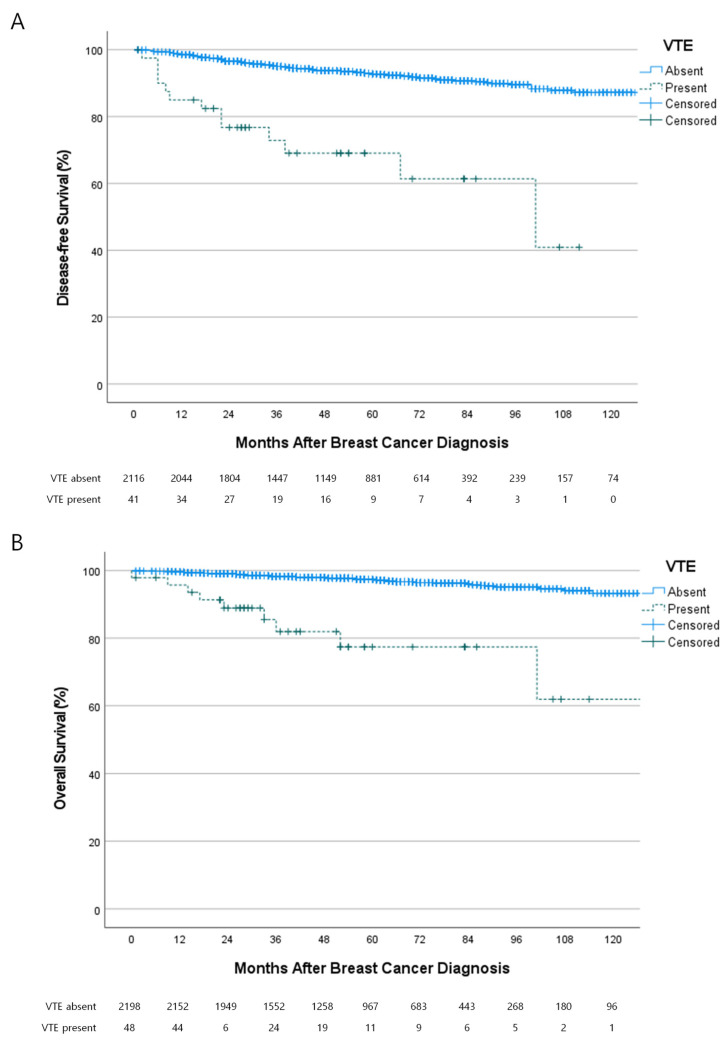
Association between the occurrence of VTE and oncologic outcomes. (**A**) VTE was associated with poor disease-free survival (Hazard ratio (HR) 6.140; 95% confidence interval, 3.480–10.835; *p <* 0.001) and (**B**) overall survival (HR 8.842; 95% CI 43.860–17.824; *p <* 0.001).

**Table 1 cancers-15-03124-t001:** Clinicopathological characteristics of the study patients.

	Patients without VTE (n = 2198)	Patients with VTE (n = 48)	*p*
Median age (range), years	51 (21–91)	61.5 (42–91)	<0.001
Age group			<0.001
≤44	505 (23.0%)	3 (6.3%)	
45–59	1170 (53.2%)	17 (35.4%)	
≥60	523 (23.8%)	28 (58.3%)	
Male sex	4 (0.2%)	2 (4.2%)	0.006
BMI, kg/m^2^	23.9 (14.1–45.4)	24.8 (18.6–35.8)	0.193
Comorbidity			0.113
Absent	1892 (86.1%)	37 (77.1%)	
Present	306 (13.9%)	11 (22.9%)	
Operation			0.002
Breast-conserving surgery	1650 (75.1%)	30 (62.5%)	
Total mastectomy	536 (24.4%)	16 (33.3%)	
Mastectomy with reconstruction	12 (0.5%)	2 (4.2%)	
Bilaterality	89 (4.0%)	3 (6.3%)	0.445
Stage			<0.001
0	327 (14.9%)	0	
I	859 (39.1%)	11 (22.9%)	
II	746 (33.9%)	23 (47.9%)	
III	222 (10.1%)	10 (20.8%)	
IV	44 (2.0%)	4 (8.3%)	
Histology			0.075
DCIS or microIDC	385 (17.5%)	1 (2.1%)	
IDC	1605 (73.0%)	41 (85.4%)	
ILC	66 (3.0%)	2 (4.2%)	
Mucinous	50 (2.3%)	0	
Medullary	15 (0.7%)	1 (2.1%)	
Metaplastic	16 (0.7%)	1 (2.1%)	
Others	61 (2.8%)	2 (4.2%)	
Hormonal receptor			0.171
Negative	526 (23.9%)	16 (33.3%)	
Positive	1672 (76.1%)	32 (66.7%)	
HER2 status			0.062
Negative	1652 (75.2%)	30 (62.5%)	
Positive	546 (24.8%)	18 (37.5%)	
Chemotherapy			<0.001
Not done	675 (30.7%)	6 (12.5%)	
Adjuvant	1376 (62.6%)	32 (66.7%)	
Neoadjuvant	105 (4.8%)	6 (12.5%)	
Palliative	42 (1.9%)	4 (8.3%)	
Endocrine treatment			0.004
Not done	499 (22.7%)	19 (39.6%)	
Tamoxifen ± OFS	1039 (47.3%)	12 (25.0%)	
Aromatase inhibitor	660 (30.0%)	17 (35.4%)	

BMI, body mass index; DCIS, ductal carcinoma in situ; microIDC, microinvasive ductal carcinoma; IDC, invasive ductal carcinoma; ILC, invasive lobular carcinoma; HER2, human epidermal growth factor receptor 2; OFS, ovarian function suppression.

**Table 2 cancers-15-03124-t002:** Factors associated with VTE occurrence.

Variables	Cumulative Incidence	Univariable Analysis		Multivariable Analysis	
		HR (95% CI)	*p*	HR (95% CI)	*p*
Age group					
≤44	0.6%	Reference		Reference	
45–59	1.4%	2.450 (0.718–8.361)	0.152	2.289 (0.669–7.825)	0.187
≥60	5.1%	9.204 (2.797–30.290)	<0.001	8.468 (2.562–27.984)	<0.001
Sex					
Female	2.1%	Reference		Reference	
Male	33.3%	20.225 (4.907–83.358)	<0.001	9.953 (2.170–45.647)	0.003
Operation					
Breast-conserving surgery	1.8%	Reference		Reference	
Simple mastectomy	2.9%	1.673 (0.912–3.069)	0.097	0.889 (0.446–1.772)	0.738
Mastectomy with reconstruction	14.3%	10.059 (2.399–42.174)	0.002	7.209 (1.478–35.167)	0.009
Stage					
0	0	0	0.954	0	0.954
I	1.3%	Reference		Reference	
II	3.0%	2.377 (1.159–4.875)	0.018	2.250 (1.085–4.667)	0.029
III	4.3%	3.617 (1.536–8.519)	0.003	3.506 (1.415–8.686)	0.007
IV	8.3%	7.367 (2.344–23.157)	0.001	5.766 (1.618–20.548)	0.007
HER2 status					
Negative	1.8%	Reference			
Positive	3.2%	1.805 (1.006–3.237)	0.048		
Chemotherapy					
Not done	0.9%	Reference			
Performed	2.7%	3.054 (1.298–7.183)	0.011		

HR, hazard ratio; CI, confidence interval.

**Table 3 cancers-15-03124-t003:** Factors associated with recurrences of breast cancer.

	Univariable Analysis		Multivariable Analysis	
	HR (95% CI)	*p*	HR (95% CI)	*p*
Age group				
≤44	Reference			
45–59	0.827 (0.575–1.191)	0.308		
≥60	0.812 (0.515–1.282)	0.372		
T stage				
T_is_	Reference		Reference	
1	6.016 (1.456–24.860)	0.013	4.582 (1.102–19.061)	0.036
2	22.697 (5.594–92.098)	<0.001	12.015 (2.907–49.665)	0.001
3	35.723 (7.827–163.046)	<0.001	15.294 (3.258–71.803)	0.001
4	67.903 (14.875–309.981)	<0.001	17.674 (3.716–84.069)	<0.001
N stage				
0	Reference		Reference	
1	2.393 (1.616–3.543)	<0.001	1.757 (1.173–2.633)	0.006
2	4.423 (2.766–7.075)	<0.001	2.341 (1.423–3.852)	0.001
3	15.062 (9.831–23.074)	<0.001	9.006 (5.696–14.238)	<0.001
Hormonal receptor				
Negative	Reference		Reference	
Positive	0.380 (0.278–0.519)	<0.001	0.377 (0.271–0.525)	<0.001
HER2 status				
Negative	Reference		Reference	
Positive	1.229 (0.875–1.727)	0.234	0.652 (0.454–0.935)	0.020
VTE				
Absent	Reference		Reference	
Present	6.140 (3.480–10.835)	<0.001	3.426 (1.909–6.149)	<0.001

HR, hazard ratio; CI, confidence interval; HER2, human epidermal growth factor receptor 2; VTE, venous thromboembolism.

**Table 4 cancers-15-03124-t004:** Factors associated with mortality.

	Univariable Analysis		Multivariable Analysis	
	HR (95% CI)	*p*	HR (95% CI)	*p*
Age group				
≤44	Reference			
45–59	1.213 (0.642–2.293)	0.552		
≥60	2.058 (1.041–4.071)	0.038		
T stage				
T_is_	Reference			
1	1.637 (0.482–5.565)	0.430		
2	4.919 (1.510–16.029)	0.008		
3	11.336 (2.835–45.339)	0.001		
4	28.803 (7.796–106.418)	<0.001		
N stage				
0	Reference		Reference	
1	0.890 (0.364–2.179)	<0.799	0.906 (0.370–2.221)	0.830
2	5.787 (2.945–11.374)	<0.001	4.032 (2.016–8.061)	<0.001
3	24.380 (14.098–42.161)	<0.001	13.495 (7.265–25.065)	<0.001
M stage				
0	Reference		Reference	
1	19.667 (11.159–34.661)	<0.001	4.971 (2.600–9.503)	<0.001
Hormonal receptor				
Negative	Reference		Reference	
Positive	0.293 (0.184–0.466)	<0.001	0.282 (0.174–0.457)	<0.001
HER2 status				
Negative	Reference			
Positive	1.780 (1.102–2.874)	0.018		
VTE				
Absent	Reference		Reference	
Present	8.842 (4.386–17.824)	<0.001	4.863 (2.367–9.989)	<0.001

HR, hazard ratio; CI, confidence interval; HER2, human epidermal growth factor receptor 2; VTE, venous thromboembolism.

**Table 5 cancers-15-03124-t005:** Clinical presentations of the VTE events.

Classification	Frequency
Presentation	
DVT	18 (37.5%)
PTE	25 (52.1%)
DVT and PTE	5 (10.4%)
Symptom	
Absent	25 (52.1%)
Present	23 (47.9%)
Timing of VTE diagnosis	
Neoadjuvant chemotherapy	4 (8.3%)
Within 1 month after surgery	3 (6.3%)
Adjuvant chemotherapy	16 (33.3%)
Follow up	17 (35.4%)
After metastasis	8 (16.7%)
Treatment	
Observation	11 (22.9%)
Unfractionated heparin	2 (4.2%)
Warfarin ± LMWH	11 (22.9%)
DOAC	24 (50.0%)

DVT, deep vein thrombosis; PTE, pulmonary thromboembolism; LMWH, low molecular weight heparin; DOAC, direct oral anticoagulant.

## Data Availability

Data are available from the authors upon request.

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
