# Peer review of "The Risk of Venous Thromboembolism in Korean Patients with Breast Cancer: A Single-Center Experience"

_cancers, 2023, doi:10.3390/cancers15123124_

Round 1

Reviewer 1 Report

In this retrospective monocentric study, the authors report the incidence of VTE among Korean patients with breast cancer and identify several risk factors.

While the data are clearly presented and discussed and bring significant conclusions, there are some points that need to be addressed to make the paper suitable for acceptance:

1/ The main limitation of this study is its retrospective design and the absence of systematic screening for VTE (line 90 page 2), whereas it is stated line 17 page 6 that 70% of PTE were asymptomatic. A subanalysis considering only symptomatic events should be performed. This limitation should also be mentioned in the discussion.

2/ I am not sure that the results obtained from a Korean population are extrapolable to the whole Asian population...

3/ In the simple summary, the authors claim a close association between VTE and breast cancer and that the incidence of VTE is not low. This should be nuanced as, even if VTE incidence is higher than in the general Korean population, the incidence compared to other cancer types is not reported here...

4/ Did the 3 patients died from VTE or from advanced cancer disease ?

5/ Page 1 line 36: I am not sure of the causative negative effect of VTE on oncologic outcome. It would be more appropriate to reformulate as "VTE is associated with a negative oncologic outcome"

6/ Page 3 line 10: do all variables present a Gaussian distribution ? Otherwise, they should not be presented as mean and sd and Student t-test is not appropriate.

7/ It is not mentioned anywhere if the study has been approved by any ethics committee...

8/ Page 3, line 12: the last 3 lines about risk factors for VTE should be presented later in the manuscript (Page 5 line 14)

9/ Page 4, line 13: the presentation of VTE timing is confusing: 32 VTE occured within 12 months but only 5 occured between 6 and 12 months... same for 24 months. This should be reformulated for a better comprehension.

10/ One main result rom this study is the identification of reconstruction as a strong risk factor for VTE. Do the authors have an hypothesis to explain this result ? In these particular cases, what is the timing of VTE compared to surgery ?

11/ Page 5 table 2: why HER2 status and Chemotherapy  were not included in the multivariate analysis ?

12/ Page 7 line 18: specific causes were identified in how many patients in total ? Cancer-related and not related should be presented separately.

13/ The authors mentioned the Khorana score in the discussion: do they have access to whole blood cell count to calculate this score in their patients ?

Author Response

In this retrospective monocentric study, the authors report the incidence of VTE among Korean patients with breast cancer and identify several risk factors.

While the data are clearly presented and discussed and bring significant conclusions, there are some points that need to be addressed to make the paper suitable for acceptance:

1/ The main limitation of this study is its retrospective design and the absence of systematic screening for VTE (line 90 page 2), whereas it is stated line 17 page 6 that 70% of PTE were asymptomatic. A subanalysis considering only symptomatic events should be performed. This limitation should also be mentioned in the discussion.

Response: We thank the reviewer for precious opinions. We additionally performed subgroup analysis for symptomatic VTE. There was no significant difference on clinicopathological characteristics between asymptomatic and symptomatic VTE except male breast cancer (Supplementary Table S1-1). Old age, male sex, operation procedures, advanced stage, and hormonal receptor were associated with symptomatic VTE (Supplementary Table S1-2). Symptomatic VTE was associated with disease-free survival and overall survival, respectively (Supplementary Table S1-3, S1-4). There was no significant difference on disease-free survival and overall survival between patients with asymptomatic VTE and symptomatic VTE (Supplementary Figure S1).

2/ I am not sure that the results obtained from a Korean population are extrapolable to the whole Asian population...

Response: We agree with the reviewer’s comment. Asians consist of heterogenous ethnic subgroups, including Chinese and Indian. The result from our study is not extrapolable to whole Asian population, and it should be validated in the multiple ethnic groups in Asia.

Nevertheless, patients with breast cancer in Asian countries have significant differences from Western countries [1]. Asians were a minor subgroup in the fields of breast cancer and venous thromboembolism. Data are scarce regarding the association between VTE and breast cancer in Asia, and our study provides new knowledge in the field of cardio-oncology in Asians.

3/ In the simple summary, the authors claim a close association between VTE and breast cancer and that the incidence of VTE is not low. This should be nuanced as, even if VTE incidence is higher than in the general Korean population, the incidence compared to other cancer types is not reported here...

Response: We thank the reviewer for a critical comment. The average incidence of VTE in our study was 459 per 100,000 person-years which was far less than that of advanced gastric cancer (1,880 per 100,000) [2]. The cumulative incidence of VTE in our study was 2.1%, which is less than that of pancreatic cancer (5.8%) [3] and non-smallcell lung cancer (6.4%) [4] in Korea.

4/ Did the 3 patients died from VTE or from advanced cancer disease ?

Response: The three patients mentioned in our manuscript died from VTE. Most of the patients with metastatic disease eventually succumb to cancer itself. However, VTE may deprive the chance of further chemotherapy, and shorten the survival of the cancer patients.

5/ Page 1 line 36: I am not sure of the causative negative effect of VTE on oncologic outcome. It would be more appropriate to reformulate as "VTE is associated with a negative oncologic outcome"

Response: We agree with the reviewer’s comment. Both VTE and oncologic outcomes of breast cancer are associated with biologic aggressiveness of the tumor [5]. VTE is associated with a negative oncologic outcome, rather than VTE directly causes negative oncologic outcome.

6/ Page 3 line 10: do all variables present a Gaussian distribution ? Otherwise, they should not be presented as mean and sd and Student t-test is not appropriate.

Response: We thank the reviewer for a critical comment. The two continuous variables (age and body mass index) did not pass the normality tests. Therefore, we changed to perform Mann-Whitney U test, instead of Student’s t-test.

7/ It is not mentioned anywhere if the study has been approved by any ethics committee...

Response: We stated ethical statement after conclusion. The study was conducted in accordance with the Declaration of Helsinki, and approved by the Institutional Review Board of Hallym University Sacred Heart Hospital (IRB File No. 2023-01-005).

8/ Page 3, line 12: the last 3 lines about risk factors for VTE should be presented later in the manuscript (Page 5 line 14)

Response: We thank the reviewer for a correction. We moved the sentences to the next paragraph.

9/ Page 4, line 13: the presentation of VTE timing is confusing: 32 VTE occured within 12 months but only 5 occured between 6 and 12 months... same for 24 months. This should be reformulated for a better comprehension.

Response: We thank the reviewer for a suggestion. To help the readers to comprehend our data well, we inserted another figure.

10/ One main result rom this study is the identification of reconstruction as a strong risk factor for VTE. Do the authors have an hypothesis to explain this result ? In these particular cases, what is the timing of VTE compared to surgery ?

Response: Long operation time risks the patients to venous pooling. Complex procedures may release cytokines which activate procoagulant pathways. Especially, autologous reconstruction is at high risk for VTE. However, only 14 (0.6%) patients were included in our study because delayed reconstruction was favored at our institution during study periods.

 Among 14 patients who underwent reconstructive procedures, there were 8 (57.1%) cases of split thickness skin graft, 3 (21.4%) cases of tissue expander insertion, and 1 (7.1%) case of latissimus dorsi myocutaneous flap, transverse rectus abdominis muscle flap, and thoracoabdominal flap, respectively.

11/ Page 5 table 2: why HER2 status and Chemotherapy  were not included in the multivariate analysis ?

Response: We thank the reviewer for a critical comment. There was an association between the two variables – chemotherapy and breast cancer stage. Patients who received chemotherapy are more likely to have advanced breast cancer, and vice versa. Advanced breast cancer stage is the cause of chemotherapy. Similarly, positive HER2 status was associated with advanced stage of breast cancer. Chemotherapy and HER2 status was not significant after multivariable analysis.

12/ Page 7 line 18: specific causes were identified in how many patients in total ? Cancer-related and not related should be presented separately.

Response: We separately discussed specific causes of venous thromboembolism. However, it is difficult to dichotomize whether VTE is related to cancer or not. The pathogenesis of VTE is multifactorial, and cancer-asssociated VTE is a heterogeneous disease entity. The mainstay of treatment for VTE in cancer patients is anticoagulation, regardless of cause.

Of the 48 patients with VTE in this study, 10 (20.8%) patients had a condition other than breast cancer, including traumatic injury (n=5), May-Thurner syndrome (n=3), and immobilized state (n=2). Rather than classified as non-cancer VTE, these patients are regarded to have multiple risk factors for developing VTE.

13/ The authors mentioned the Khorana score in the discussion: do they have access to whole blood cell count to calculate this score in their patients ?

Response: We thank the review for a suggestion. In a pilot study, we planned to validate Khorana score in the patients with metastatic breast cancer. However, most of the patients presented with normal blood cell count before chemotherapy. It would be difficult to find the usefulness of the Khorana score in our study.

References

  1. Leong, S.P.; Shen, Z.-Z.; Liu, T.-J.; Agarwal, G.; Tajima, T.; Paik, N.-S.; Sandelin, K.; Derossis, A.; Cody, H.; Foulkes, W.D. Is breast cancer the same disease in Asian and Western countries? World journal of surgery 2010, 34, 2308-2324.
  2. Kang, M.J.; Ryoo, B.-Y.; Ryu, M.-H.; Koo, D.-H.; Chang, H.M.; Lee, J.-L.; Kim, T.W.; Kang, Y.-K. Venous thromboembolism (VTE) in patients with advanced gastric cancer: an Asian experience. European Journal of Cancer 2012, 48, 492-500.
  3. Oh, S.Y.; Kim, J.H.; Lee, K.-W.; Bang, S.-M.; Hwang, J.-H.; Oh, D.; Lee, J.S. Venous thromboembolism in patients with pancreatic adenocarcinoma: lower incidence in Asian ethnicity. Thromb Res 2008, 122, 485-490.
  4. Lee, Y.-G.; Kim, I.; Lee, E.; Bang, S.-M.; Kang, C.H.; Kim, Y.T.; Kim, H.J.; Wu, H.-G.; Kim, Y.W.; Kim, T.M. Risk factors and prognostic impact of venous thromboembolism in Asian patients with non-small cell lung cancer. Thrombosis and haemostasis 2014, 111, 1112-1120.
  5. Mahe, I.; Scotte, F.; Elalamy, I. Tumor Genetics Are Thrombogenic: The Need for Action. JACC CardioOncol 2023, 5, 256-258, doi:10.1016/j.jaccao.2023.03.002.

Reviewer 2 Report

Authors report an intrighino retrospective analysis on a well known topic for cancer associate thrombosis.

results are intriguing but i well ad a further table in which a differentiation between inpatients and outpatients is present and among patients on treatment (e.g. chemotherapy alone, chemotherapy and radiotherapy, chemotherapy and anti hormonal therapy) and off anti cancer treatment are specified.

conclusion should also include a more detailed suggestion to prevent this kind of cancer associated thrombosis

Author Response

Authors report an intrighino retrospective analysis on a well known topic for cancer associate thrombosis.

results are intriguing but i well ad a further table in which a differentiation between inpatients and outpatients is present and among patients on treatment (e.g. chemotherapy alone, chemotherapy and radiotherapy, chemotherapy and anti hormonal therapy) and off anti cancer treatment are specified.

conclusion should also include a more detailed suggestion to prevent this kind of cancer associated thrombosis

Responses: We thank the reviewer for suggestions. In our institutions, most of the patients received chemotherapy after admission. Discriminating inpatients from outpatients may not be valuable in our study.

Patients in our study received anti-cancer treatment sequentially. We did not offer chemotherapy and radiotherapy, or chemotherapy and anti-hormonal therapy concurrently. For those who underwent upfront surgery, radiation treatment was given three weeks after chemotherapy had been finished.

As we described in Table 3, four (8.3%) cases of VTE occurred during neoadjuvant chemotherapy, three (6.3%) occurred after surgery, 16 (33.3%) occurred during adjuvant chemotherapy, and 17 (35.4%) occurred during follow-up, and eight (16.7%) occurred after metastasis.

Because the occurrence of VTE in breast cancer is still low, we do not recommend routine anticoagulation for most of the patients with breast cancer. Rather, we suggest an individualized approach by balancing the risk of bleeding and thrombosis.

Reviewer 3 Report

The manuscript by Jung Ho Park et al., titled “The risk of venous thromboembolism in Korean patients with 2 breast cancer: A single center experience” is a single centre study which examines risk factors for venous thromboembolism in Korean patients with breast cancer. The study is well-designed and well-performed, and I believe it will be useful for the scientific community if it is published. I only have a few comments to make, as follows:

1.       The study includes 2,246 patients, so it is not appropriate to extend the applicability of it to the Korean population. Therefore, I suggest that the authors change the phrase “in the Korean population” in line 70 to “in a Korean cohort”.

2.       There were only 6 male patients in the study population (4 without VTE and 2 with VTE). Observing the difference (and a reported p value of 0.006) the authors have deducted that “male breast cancer was a strong risk factor for VTE in the present study”. Given the small number of male patients, the abovementioned difference could simply have been caused by chance. How the authors have high confidence that this is not caused by chance?

3.       In terms of assessing the association of comorbidities with VTE, in table 1 there is only one category called comorbidities. We all know there are many types of comorbidities (such as diabetes mellitus, atrial fibrillation, heart failure, etc.). My question is that whether the authors did sub-group analyses for major groups of comorbidities and if they found any associations? If not, what was the reason the authors put all types of comorbidities in one group?

Author Response

The manuscript by Jung Ho Park et al., titled “The risk of venous thromboembolism in Korean patients with 2 breast cancer: A single center experience” is a single centre study which examines risk factors for venous thromboembolism in Korean patients with breast cancer. The study is well-designed and well-performed, and I believe it will be useful for the scientific community if it is published. I only have a few comments to make, as follows:

  1. The study includes 2,246 patients, so it is not appropriate to extend the applicability of it to the Korean population. Therefore, I suggest that the authors change the phrase “in the Korean population” in line 70 to “in a Korean cohort”.

Response: We thank the reviewer for reviewing our manuscript and giving a good suggestion. We changed the phrase ”in the Korean population” to “in a Korean cohort”.

  1. There were only 6 male patients in the study population (4 without VTE and 2 with VTE). Observing the difference (and a reported p value of 0.006) the authors have deducted that “male breast cancer was a strong risk factor for VTE in the present study”. Given the small number of male patients, the abovementioned difference could simply have been caused by chance. How the authors have high confidence that this is not caused by chance?

Response: We totally agree with the reviewer’s comment. Because the number of male breast cancers was small, the association between male sex and VTE should be carefully interpreted. Although the “p value” of male patients was significant in our study, male sex is not an established risk factor for VTE. The association between male sex and VTE should be further evaluated. Also, we did not exclude male patients in our study because this would not change our conclusion.

  1. In terms of assessing the association of comorbidities with VTE, in table 1 there is only one category called comorbidities. We all know there are many types of comorbidities (such as diabetes mellitus, atrial fibrillation, heart failure, etc.). My question is that whether the authors did sub-group analyses for major groups of comorbidities and if they found any associations? If not, what was the reason the authors put all types of comorbidities in one group?

Response: Thank the reviewer for giving a critical point. We performed a subgroup analysis as to comorbidities. VTE was associated with comorbidities such as hypertension, diabetes mellitus, and chronic kidney disease (Supplementary Table S1). However, we could not identify significant association between VTE and infrequent comorbidities (e.g. atrial fibrillation, heart failure) in our study.

 In the Cox proportional hazard model, chronic kidney disease was an independent risk factor for developing VTE (odds ratio, 5.742; 95% confidence interval 1.298–25.404; p=0.025). Because comorbidities such as hypertension and diabetes mellitus were associated with old age, they were not statistically significant on multivariable analysis.

Round 2

Reviewer 1 Report

The authors replied to most of my comments.

Only one question remained unanswered: The authors claim that breast reconstruction was a strong risk factor for VTE. However, only 2 patients developped VTE after breast reconstruction: what was the timing of VTE ? Did VTE occur 1-day, 1-week, 1-month after ? It is a crucial information to attribute VTE to the reconstruction procedure...

Author Response

Only one question remained unanswered: The authors claim that breast reconstruction was a strong risk factor for VTE. However, only 2 patients developped VTE after breast reconstruction: what was the timing of VTE ? Did VTE occur 1-day, 1-week, 1-month after ? It is a crucial information to attribute VTE to the reconstruction procedure...

Response: We reported a high incidence of VTE among patients who underwent reconstructive procedures (2/14=14.3%). Because the number of patients with VTE was small, this unexpectedly high incidence of VTE may be partially caused by chance, as the reviewer pointed out.

  One patient was a 91-year-old woman who had a history of hypertension. She presented with an ulcerating mass on her right breast. She underwent modified radical mastectomy with wide skin excision. The skin defect was covered with full-thickness skin graft. She developed with bilateral lower leg edema 9 days after surgery, and was diagnosed with bilateral deep vein thrombosis.

  The other patient was an 84-year-old woman who had hypertension, asthma, and Alzheimer’s disease. She presented with an 8cm extent Paget’s disease. An extensive skin excision was performed, requiring reconstruction with split-thickness skin graft. A 2.8-cm sized invasive breast cancer was identified after surgery. After two months, she slipped down and presented with femur neck fracture on her right side. She underwent hemiarthroplasty, and was discharged without an event. She developed bilateral leg swelling after four months of breast cancer diagnosis, and was diagnosed with bilateral deep vein thrombosis and pulmonary thromboembolism.

  The two patients do not represent the whole patients who undergo breast reconstruction. We also agree that implant-based breast reconstruction per se does not elevate the risk of VTE. However, those patients have similar characteristics such as old age, advanced cancer stage, and reconstructive procedures.

  The cause of VTE is multifactorial, and it is difficult to fully differentiate between cancer-associated VTE and non-cancer VTE. It is evident that breast cancer contributed to the pathogenesis of VTE. Excluding this subgroup of patients would cause selection bias.
